

# Effectiveness, tolerability and safety of Direct Acting Antivirals in Mexican individuals with Hepatitis C virus genotype-1 and previous pegylated interferon and ribavirin therapy

Daniel Melendez-Mena[1,2], Miguel Angel Mendoza-Torres[1,2], Virginia Sedeño-Monge[3], Víctor Hugo García y García[2], Elain Rivera-García[3], Laura Sánchez-Reza[2], María del Carmen Baxin Domínguez[4], Belinda Guzmán-Flores[5], Ygnacio Martinez-Laguna[6], José Manuel Coronel Espinoza[7], Iván Galindo-Santiago[8], Juan Carlos Flores-Alonso[8], Verónica Vallejo-Ruiz[8], Paulina Cortes-Hernandez[9], Julio Reyes-Leyva[8], Francisca Sosa-Jurado[8] and Gerardo Santos-López[8]

[1] Centro Interdisciplinario de Posgrados, Facultad de Medicina, Universidad Popular Autonóma del Estado de Puebla, Puebla, Puebla, Mexico
[2] Servicio de Gastroenterología, Centro Médico Nacional General de División Manuel Ávila Camacho, Instituto Mexicano del Seguro Social, Puebla, Puebla, Mexico
[3] Decanato de Ciencias Médicas, Universidad Popular Autonóma del Estado de Puebla, Puebla, Puebla, Mexico
[4] Centro Médico Nacional La Raza, Instituto Mexicano del Seguro Social, Ciudad de México, Mexico
[5] Banco de Sangre, Centro Médico Nacional General de División Manuel Ávila Camacho, Instituto Mexicano del Seguro Social, Puebla, Puebla, Mexico
[6] Instituto de Ciencias, Benemerita Universidad Autónoma de Puebla, Puebla, Puebla, Mexico
[7] Hospital General Regional # 1, Instituto Mexicano del Seguro Social, Tarímbaro, Michoacán, Mexico
[8] Laboratorio de Biología Molecular y Virología, Centro de Investigación Biomédica de Oriente, Instituto Mexicano del Seguro Social, Metepec, Puebla, Mexico
[9] Laboratorio de Biología Celular, Centro de Investigación Biomédica de Oriente, Instituto Mexicano del Seguro Social, Metepec, Puebla, Mexico

Corresponding author
Gerardo Santos-López,
gerardo.santos.lopez@gmail.com

## ABSTRACT

**Background.** Direct Acting Antivirals (DAAs) represent a large improvement in the treatment of chronic hepatitis C, resulting in <90% sustained virological response (SVR). There are no reports on the real-world DAA response for Mexico and few reports exist for Latin America. The aim of the study was to report SVR, and immediate benefits with the DAA treatments sofosbuvir, ledispavir, with/without ribavirin (SOF/LDV ± RBV) and ombitasvir, paritaprevir, ritonavir, dasabuvir with/without RBV (OBV/PTV/r/DSV ± RBV) in patients with viral genotype 1a or 1b, and who did not respond to previous peginterferon/ribavirin (PegIFNα2a+RBV) therapy.

**Methods.** A descriptive, ambispective, longitudinal study was conducted. A cohort of 261 adult patients received PegIFNα2a+RBV therapy before 2014; 167 (64%) did not respond, 83 of these were subsequently treated with SOF/LDV ± RBV or OBV/PTV/r/DSV ± RBV. Child-Pugh-Score (CPS), Fibrosis-4 (FIB-4), and AST to Platelet Ratio Index (APRI) were evaluated before and after treatment.

**Results**. SVR with PegIFNα2a+RBV was 36%, and 97.5% with DAAs. CPS, FIB-4 and APRI improved significantly after DAA treatment, mainly because of liver transaminase reduction.

**Conclusions**. DAA treatment showed excellent SVR rates in Mexican patients who had not responded to PegIFNα2a+RBV therapy. Improvement in CPS, FIB-4 and APRI without improvement in fibrosis was observed in cirrhotic and non-cirrhotic patients, as well as considerable reduction in liver transaminases, which suggests a reduction in hepatic necroinflammation.

## INTRODUCTION

In 2015, the World Health Organization (WHO) estimated 1.75 million new hepatitis C virus (HCV) infections and that 71 million people in the world were living with chronic hepatitis C (CHC). Approximately 400,000 people died from CHC related liver diseases mainly cirrhosis and hepatocellular carcinoma (HCC) in that year (*World Health Organization, 2017*).

For nearly a decade, the first-line therapy against CHC was pegylated interferon alpha (pegIFNα) with ribavirin (RBV), but sustained virological response (SVR) was observed only in 40–50% of patients infected with viral genotype 1 and in 70% of those with genotypes 2 or 3 (*Hofmann et al., 2015*). In the last five years, the direct-acting antiviral agents (DAAs), have revolutionized CHC treatment (*Elbaz, El-Kassas & Esmat, 2015*). The first generation DAAs (boceprevir and telaprevir) were serine protease inhibitors used in triple combination with interferon and RBV that had low tolerability (*Watanabe et al., 2016*). New DAA combinations without interferon (*i.e.,* sofosbuvir, simeprevir and ombitasvir), that target several viral-cycle proteins, like NS3/4A (protease), NS5A and NS5B (RNA polymerase), were introduced in 2013–2014 and show enhanced efficacy and tolerability. These second generation DAAs, have increased SVR rates to over 90% and effectively treat all HCV genotypes (*Gotte & Feld, 2016*) even in patients with advanced cirrhosis (*Mizokami et al., 2015*), making them efficient curative therapies. However, access to DAAs varies greatly among countries. According to WHO, only around half of patients with CHC that started treatment in 2015 received DAAs, and they concentrated in high income countries (*World Health Organization, 2017*). Thus, data on real-world safety and effectiveness of DAA regimens are still emerging for the middle- and low-income countries that harbor most of the CHC patients. There is limited data on real-world DAA treatments administered through public health systems in Latin America (LA), which are crucial in the efforts to eradicate HCV. Delivering DAA treatments to patients in need is now a main challenge worldwide. The high costs of DAA regimens are restrictive even for patients in high-income countries. This results in a portion of patients being treated only after liver

fibrosis/cirrhosis have emerged, and highlights the need to understand the effects of DAAs on liver fibrosis and inflammation.

In Mexico the most frequent HCV genotype is 1. Treatment for this genotype with pegIFNα+RBV has a low SVR rate of around 38% (*Sandoval-Ramirez et al., 2015*), while the SVR with DAAs has not been reported in this country and few reports exist for LA (*Cheinquer et al., 2017*). Large public health systems such as the Mexican Institute of Social Security (IMSS) that currently tends to ≈60% of the Mexican population (*Instituto Mexicano del Seguro Social, 2018*), introduced DAA regimens in 2017, initially treating patients who received and did not respond to pegIFNα2a+RBV. We studied Mexican adults with CHC that were treated with pegIFNα2a+RBV between 2005 and 2014. The non-responders from that cohort were among the first treated with DAAs in a real-world, public health setting in Mexico between 2017 and 2019 and their results are reported here.

## MATERIALS & METHODS

### Study design
We conducted a descriptive, ambispective and longitudinal study in two treatment phases for patients with chronic HCV infection on a cohort of patients who initially received pegIFNα2a+RBV treatment before 2014; non-responding patients were subsequently treated with DAAs. Tolerability, safety and effectiveness were analyzed.

### Initial cohort with pegIFNα2a+RBV treatment (2005 to 2014)
The cohort study was carried out with 261 patients at a High Specialty Medical Unit of the Mexican Institute of Social Security (UMAE-HE, IMSS) in the city of Puebla, Mexico. All patients were 18 years old or over, with CHC, infected by HCV genotype 1 (1a, 1b, 1a1b), without cirrhosis or with compensated cirrhosis.

### HCV genotype and viral load in the initial cohort
The viral genotype was determined with INNO-LiPA HCV II (Inogenetics, Zwijnarde, Belgium), and the viral load with AmpliPrep/Cobas and AmpliPrep/Cobas TaqMan HCV test (Roche Molecular Systems, Indianapolis, IN, USA), before treatment and at 4, 12, 24, 48 and 72 weeks after the start of pegIFNα+RBV.

### pegIFNα2a+RBV Treatment
Patients received standard therapy for 48 weeks with pegIFNα2a 180 μg/week plus ribavirin, at a dose adjusted by body weight in the range of 1,000–1,200 mg/day. Rapid virological response (RVR, defined as undetectable HCV RNA after 4 weeks of treatment), complete early virological response (cEVR, undetectable HCV RNA after 12 weeks of treatment), partial early virological response (pEVR, a $\geq$ 2-log10 decrease in HCV RNA after 12 weeks of treatment), slow virological response (SLVR, a $\geq$ 2-log10 decrease in HCV RNA after 12 weeks and undetectable after 24 weeks of treatment), null response (decrease in HCV RNA <2 logs at week 12), SVR (absence of detectable HCV-RNA at 24 weeks after the end of treatment), relapse (HCV-RNA became undetectable during treatment but reappeared after treatment discontinuation), breakthrough (detection of HCV RNA at any point

during treatment after dropping to undetectable levels or an increase >1 log compared to the nadir); non-responder (NR) includes null response, relapse and breakthrough patients, to pegIFNα2a+RBV were determined according to the Mexican consensus on the diagnosis and management of hepatitis C infection (*Sanchez-Avila et al., 2015*).

## Sub-cohort of non-responder patients to pegIFNα2a+RBV that underwent DAA treatment in 2017 to 2019

A sub-cohort of patients classified as NR to pegIFNα2a+RBV that returned for, and completed, DAA treatment between June 2017 and January 2019 was studied. These patients were candidates for DAA treatment irrespective of their cirrhosis degree.

## Laboratory and imaging tests in the sub-cohort with DAA treatment

Cirrhosis was diagnosed with clinical, and laboratory data, plus hepatic elastography and ultrasound. A single hepatic elastography performed pre-treatment (FibroScan 530 Compact, Echosens, France) was used to determine steatosis, and METAVIR score of Fibrosis. Serum determinations of total bilirubin, albumin, International Normalization Ratio of prothrombin time (INR), aspartate aminotransferase (AST), alanine aminotransferase (ALT), platelet counts (PC), the presence of ascites, or hepatic encephalopathy, Child-Pugh-Score (CPS), Fibrosis-4 (FIB-4), and AST to Platelet Ratio Index (APRI) were evaluated before and after treatment.

Confirmation of the viral genotype, and HCV RNA viral load determinations were done with the Real Time System HCV Assay (Abbott Molecular, Abbott Park, Illinois USA). The lower limit of quantification (LLOQ) is 1.39 $Log_{10}$ IU/mL, and lower limit of detection (LLOD) is 1.08 $Log_{10}$IU/ml. The viral load was measured at DAA treatment start, at 12 weeks (treatment end) and at 24 weeks (12 weeks after treatment end).

## Treatment with *OBV/PTV/r/DSV* ± RBV or SOF/LDV ± RBV

Patients were evaluated by the group of experts in the management of hepatitis C (GEMHEC) at IMSS, who determined which of two available DAA regimens was best for each patient, based on medical criteria: Patients without cirrhosis, or with cirrhosis CPS-A to CPS-B, were treated with *OBV/PTV/r/DSV* ± *RBV* at 25/150/100/500 ± 1000–1200 mg per day, for 12 weeks. Patients, without cirrhosis, or with cirrhosis CPS-A to CPS-C, were treated with *SOF/LDV* ± *RBV* at 400/90 ± 1000–1200 mg per day, for 12 weeks. Patients with decompensated cirrhosis received RBV in their treatment regimen. The effectiveness of each regime was assessed by the percentage of patients with SVR12 defined as undetectable plasma HCV RNA 12 weeks after treatment end.

## Ethical aspects

The study was performed in accordance with ethical regulations and approved by the committee of research and ethics (Local Committee for Health Research No. 2101), IMSS (Registry numbers R-2004-2101-008, R-2008-2101-10, and R-2019-2101-001). Written informed consent was obtained from all patients prior to entering the study.

**Table 1** Demographic and clinical characteristics of the cohort of 261 patients with CHC, viral genotype 1, that received treatment with pegIFNα2a+RBV between 2005 and 2014.

| | n (%) |
|---|---|
| *Sex* | |
| Women /Men | 165/96 (63.2/36.7) |
| *Age* | |
| Range (years) | 15–73 |
| <50 (year old)/≥50 (year old) | 128/133 (49/51) |
| *Mean age (years) (95% CI)* | |
| Women/Men | 50 (48.5–52.4)/45 (42.8–47.4) |
| *Diagnosis* | |
| Non cirrhosis /Cirrhosis | 159/102 (61/39) |
| *Comorbidity* | |
| Diabetes mellitus type 2 | 35 (13.4) |
| Hemophiliac | 3 (1.14) |
| HCC during study | 6 (2.39) |
| HIV | 2 (0.76) |
| HBV | 3(1.14) |
| *Viral genotypes* | |
| 1 (unsubtyped) | 10 (3.8) |
| 1a | 110 (42.1) |
| 1b | 121 (46.4) |
| 1a1b | 20 (7.7) |
| Total genotype 1 | 261 (100.0) |
| *Basal viral load range ($Log_{10}$ IU/ml)* | |
| Genotype 1 | 1.96–7.25 |
| *Mean basal viral load ($Log_{10}$ IU/ml) (95% CI)* | |
| Genotype 1 | 5.59 (5.50–5.68) |

## Statistical analysis

For quantitative variables, means (95% CI), paired and unpaired Student's t test were calculated. For qualitative variables, frequencies, percentages, univariate analysis with chi-square or Fisher's exact tests, and multivariate analysis with multinomial logistic regression, were performed. Statistical significance was defined as $p \leq 0.05$. All statistical analyses were done with GraphPad Prism version 5.0 (GraphPad software, Inc. San Diego CA).

## RESULTS

### Response to pegIFNα2a+RBV treatment

We analyzed the response of 261 Mexican adults with CHC and viral genotype 1, with or without cirrhosis, who received treatment with pegIFNα2a+RBV between 2005 to 2014 (patient characteristics in Table 1). 36% of patients (94) achieved SVR, while 64% (167) were non-responders (Table 2).

**Table 2  Final response to pegIFNα2a+RBV or DAA treatments.**

**pegIFNα2a+RBV treatment of 261 Mexican patients with CHC and viral genotype 1**

| Final response | Subtypes of Genotype 1 | | | |
| | 1 + 1a1b[*] | 1a | 1b | Total |
| --- | --- | --- | --- | --- |
| NR, n (%) | 18 (6.9) | 69 (26.4) | 80 (30.7) | 167 (64) |
| SVR, n (%) | 12 (4.6) | 41 (15.7) | 41 (15.7) | 94 (36) |
| Total | 30 (11.5) | 110 (42.1) | 121 (46.4) | 261 (100) |

**DAA treatments in 83 previously treated Mexican patients with CHC and viral genotype 1**

| Regimen | SVR12, n (%) | NR12, n (%) | Total |
| --- | --- | --- | --- |
| OBV/PTV/r/DSV+RBV | 7 (87.5) | 1 (12.5) | 8 (100) |
| OBV/PTV/r/DSV | 30 (96.8) | 1 (3.2) | 31 (100) |
| Total | 37 (94.8) | 2 (5.2) | 39 (100) |
| SOF/LDV+RBV | 34 (100.0) | 0 (0.0) | 34 (100.0) |
| SOF/LDV | 10 (100) | 0 (0.0) | 10 (100) |
| Total | 44 (100) | 0 (0.0) | 44 (100) |

**Notes.**
NR, Non-responder; SVR, sustained virological response; p, value for Fisher's Exact test.
Statistical significance was defined as $p < 0.05$ (final response among genotypes).
*Includes patients with unsubtyped HCV genotype 1 ($n = 10$) and those detected as 1a1b ($n = 20$).
SVR12, sustained virological response for DAA treatments; NR12, Non-responder for DAA treatments.

Two independent variables were associated significantly with SVR after pegIFNα2a+RBV treatment in a multivariate analysis: not having cirrhosis (RR = 3.0) and having a baseline viral load <5.69 $\log_{10}$ IU/ml (RR = 3.5); the opposite conditions were associated to null response (Table 3).

pEVR was associated to SVR, but the other intermediate responses (RVR and cEVR) were ambivalently associated with SVR and relapse at the same time during pegIFNα2a+RBV treatment, so we do not consider their association to be relevant for this study (Table 3).

All of the non-responders to pegIFNα2a+RBV that were alive in 2017 were invited to receive DAA treatment and 83 patients attended. The rest (84 patients) did not receive subsequent DAA treatment through IMSS: 13 had died before 2015 and the rest were unavailable for follow-up. The baseline characteristics of the patients that received and did not receive DAAs are contrasted in Table S1.

## Sub-cohort of non-responders to pegIFNα2a+RBV that received DAA treatment

Of the 83 patients that returned for DAA treatment, 31 received OBV/PTV/r/DSV and 8 OBV/PTV/r/DSV ± RBV, while 35 received SOF/LDV ± RBV and 9 SOF/LDV, for 12 weeks. Demographic data, comorbidities, type of response to the previous treatment, steatosis grade, and basal platelet counts, were similar between the two groups (Table 4).

Viral genotype 1a predominated in the SOF/LDV ± RBV group. This group also had a lower viral load and a higher proportion of patients with cirrhosis pre-DAA treatment than the OBV/PTV/r/DSV ± RBV group (Table 4). An overall SVR rate of 97.6% was obtained at 24 weeks (12 weeks after treatment end) in 83 patients who had either of the DAA treatments, with no significant difference in SVR between treatments ($p = 0.217$) (Table 2).

**Table 3** Multivariate analysis of factors that predisposed to sustained virological response, null response, or relapse with pegIFNα2a+RBV treatment, in a cohort of 261 Mexican patients with CHC and viral genotype 1.

| Factor | SVR patients | | Null response patients | | Relapse patients | |
|---|---|---|---|---|---|---|
| | RR (95% CI) | $p$ | RR (95% CI) | $p$ | RR (95% CI) | $p$ |
| *Sex* | | | | | | |
|   Men | 1.1 (0.8–1.5) | 0.592 | 0.9 (0.7–1.3) | 0.797 | 07 (0.4–1.3) | 0.328 |
|   Women | | | | | | |
| Age (yr.) | | | | | | |
|   <50 | 1.5 (1.0–2.2) | 0.045 | 0.8 (0.6–1.0) | 0.060 | 0.5 (0.3–0.9) | 0.02 |
|   ≥50 | 1.3 (1.0–1.9) | 0.066 | 0.70 (0.4–1.4) | 0.044 | 0.5 (0.3–0.9) | 0.05 |
| *Diabetes mellitus* | | | | | | |
|   Yes | 0.9 (0.6–1.5) | 0.850 | 0.9 (0.5–1.5) | 0.737 | 1.6 (0.9–2.9) | 0.130 |
|   No | | | | | | |
| *Liver disease* | | | | | | |
|   No cirrhosis | 3.0 (1.4–6.5) | 0.002 | 1.0 (0.7–1.3) | 1.00 | 0.5 (0.3–0.9) | 0.020 |
|   Cirrhosis | 0.4 (0.3–0.7) | 0.001 | 3.5 (1.8–6.6) | <0.001 | 0.6 (0.4–1.1) | 0.110 |
| *Initial Viral load* (log$_{10}$ IU/ml) | | | | | | |
|   <5.69 | 3.5 (1.7–7.1) | 0.001 | 0.6 (0.4–0.8) | 0.003 | 0.8 (0.5–1.4) | 0.529 |
|   ≥5.69 | 0.58 (0.43–0.82 | 0.001 | 2.4 (1.3–4.3) | 0.005 | 1.8 (0.9–3.8) | 0.090 |
| *RVR* | | | | | | |
|   Yes | 2.4 (1.8–3.2) | <0.001 | NA | NA | 2.2 (1.2–4.2) | 0.020 |
|   No | | | | | | |
| *cEVR* | | | | | | |
|   Yes | 2.2 (1.6–2.9) | <0.001 | NA | NA | 5.0 (2.8–9.0) | <0.001 |
|   No | | | | | | |
| *pEVR* | | | | | | |
|   Yes | 2.4 (1.8–3.3) | 0.001 | NA | NA | 1.4 (0.6–3.6) | 0.507 |
|   No | | | | | | |

**Notes.**
NA, does not apply.
The mean viral load before pegIFNα2a+RBV was 5.69 log$_{10}$ IU/ml.
RVR, rapid virological response; cEVR, complete early virological response; pEVR, partial early virological response.

Two patients, both treated with *OBV/PTV/r/DSV* ± *RBV*, had non-response: 1 had null response, and 1 had relapse. Both patients were men, without comorbidities, and had an initial viral load >5.58 Log$_{10}$ UI/ml (Table 5). No demographic, clinical, or laboratory characteristics were significantly associated with non-response (Table S2).

At week 12 (end of treatment), 5 patients treated with OBV/PT/r/DSV ± RBV and 3 patients with SOF/LDV ± RBV still had detectable viral RNA (above LLOD but, below LLOQ). Twelve weeks after the end of treatment, 7 of these patients had undetectable viral RNA, while the last patient (treated with OBV/PTV/r/DSV+RBV) attianed undetectable viral RNA at week 24 after the end of treatment. Therefore, all 8 patients eventually achieved SVR with the DAAs. Table 5 shows their characteristics.

**Table 4  Demographic and clinical data of the Mexican patients treated with *OBV/PTV/r/DSV ± RBV* or *SOF/LDV± RBV*.**

| | Patients treated with OBV/PTV/r/DSV ± RBV (39) | Patients treated with SOF/LDV ± RBV (44) | p |
|---|---|---|---|
| *Gender, n (%)* | | | |
| Female/male | 29 (74.3)/10 (25.7) | 33 (75)/11 (25) | 0.182 |
| *Mean age* (95% CI) | 54.5(49–58) | 54.7(49.5–58) | |
| Female/male age | 57.2 (53–61)/51.8 (45–58) | 58.5 (55–62)/50 (44–55) | 0.161/0.011 |
| *Patients ≥ 50 years old, n (%)* | | | |
| All patients | 34 (87.1) | 36 (81.1) | 0.558 |
| *Response to pegIFNα2a+RBV treatment, n (%)* | | | |
| Null response | 27 (69.3) | 29 (65.9) | 0.816 |
| Relapse | 9 (23.0) | 14 (31.8) | 0.461 |
| Breakthrough | 3 (7.7) | 1 (2.3) | 0.337 |
| Total | 39 (100) | 44 (100) | |
| *Comorbidity, n (%)* | | | |
| Diabetes | 5 (12.8) | 4 (9.0) | 0.728 |
| Smoking | 3 (7.7) | 8 (18.8) | 0.204 |
| Alcohol use disorder | 5 (12.8) | 3 (6.8) | 0.465 |
| Obesity[*] | 7 (17.9) | 12 (27.3) | 0.433 |
| *Viral Subtype 1, n (%)* | | | |
| 1a | 8 (21.5) | 34 (77.3) | 0.024 |
| 1b | 31 (79.5) | 10 (22.7) | 0.024 |
| Total | 39 (100) | 44 (100) | |
| *Initial viral load ($\log_{10}$ IU/ml)* | | | |
| Mean (CI 95%) | 5.58 (5.3–5.8) | 5.10 (4.78–5.40) | 0.031 |
| *Initial viral load >5.58 ($Log_{10}$ UI/ml), n (%)* | | | |
| Yes | 15 (38.4) | 20 (45.5) | 0.656 |
| No | 24 (61.6) | 24 (54.5) | |
| Total | 39 (100) | 44 (100) | |
| *METAVIR Score[**], n (%)* | | | |
| F0 or F1 | 17 (43.6) | 8 (18.2) | 0.01 |
| F2 | 1 (2.6) | 0 (0.0) | 1.0 |
| F3 | 3 (7.7) | 4 (9.0) | 1.0 |
| F4 | 18 (46.1) | 32 (72.7) | <0.001 |
| Total | 39 (100) | 44(100.0) | |
| *Degree of steatosis[***], n (%)* | | | |
| None | 16 (41.0) | 21 (47.8) | 0.658 |
| 1 to 3 | 23 (59.0) | 10 (52.2) | 0.01 |
| Total | 39 (100) | 44 (100) | |
| *Basal albumin >3.5 g/dL, n (%)* | | | |
| Yes | 31 (79) | 21 (47.8) | 0.003 |
| No | 8 (21) | 23 (52.2) | |

**Table 4** (*continued*)

| | Patients treated with OBV/PTV/r/DSV ± RBV (39) | Patients treated with SOF/LDV ± RBV (44) | p |
|---|---|---|---|
| Total | 39 (100) | 44 (100) | |
| *Basal total bilirubin <2.0 mg/dL, n (%)* | | | |
| Yes | 35 (89) | 30 (68) | 0.03 |
| No | 4 (11) | 14 (32) | |
| Total | 39 (100) | 44 (100) | |
| *Basal platelet count >100,000/(mm3), n (%)* | | | |
| Yes | 28 (72) | 23 (52.2) | 0.001 |
| No | 11 (28) | 21 (47.8) | |
| Total | 39 (100) | 44 100) | |

**Notes.**
\*Body Mass Index (BMI) ≥30, includes obesity degrees 1–3.
\*\*METAVIR Score from Hepatic elastography (FibroScan) before DAA treatment, classified from the measurement in kPa: <7.6kPa = F0-F1, 7.7–9.4 kPa = F2, 9.5–12 kPa = F3, >12 kPa = F4.
\*\*\*Degree of steatosis (FibroScan) ≤220 dB/m = non steatosis, >220–235 dB/m = 1, 236–290 dB/m=2, >290 dB/m = 3.

## Child-Pugh Score, serological fibrosis markers FIB-4 and APRI before and after DAA treatments

CPS post-treatment improved significantly in 12 patients with cirrhosis. After OBV/PTV/r/DSV ± RBV treatment, 1 patient improved from CPS-B to -A; while after SOF/LDV ± RBV treatment, 11 patients improved: one from CPS-C to -A, and the other 10 from CPS-B to -A (Table 6).

Since hepatic elastography was carried out only pre-treatment we did not have a direct METAVIR comparison of the hepatic fibrosis before and after DAAs. Therefore, we determined the FIB-4 score, and the APRI index, to explore whether liver fibrosis changed with treatment. We found a decrease in mean FIB-4 and APRI values with both DAA regimens that was observed in both cirrhotic and non-cirrhotic patients. This decrease was enough to rate below the threshold for F4 cirrhosis in patients after *OBV/PTV/r/DSU ± RBV* but not after *SOF/LDV ± RBV,* which had more F4 cirrhosis before treatment. Non-cirrhotic patients still rated as having persistent liver fibrosis after DAAs, despite the decrease in mean FIB-4 and APRI values (Table 6).

## Tolerability and adverse events to DAAs

The adverse effects during DAA treatments were epigastralgia (18%), headache (12%), hyperbilirubinemia without elevation of ALT or AST during the first two weeks (12%), fever (2.4%), and pruritus (1.2%). All the events were tolerated and controllable and none of the patients discontinued treatment. One patient, treated with SOF/LDV ± RBV, had a sudden hepatic decompensation, that could be associated with frequent ingestion of *Peumus boldus* leave infusions during treatment. Herb-drug interactions have been documented for *P. boldus* with other drugs (*Awortwe et al., 2018*). The patient discontinued the infusions, corrected the hepatic decompensation, completed treatment and reached SVR.

Melendez-Mena et al. (2021), *PeerJ*, DOI 10.7717/peerj.12051

**Table 5** Types of response and characteristics of the patients that either failed DAA-treatment (first two rows) or had detectable viral RNA at the end of DAA treatment (week 12) but finally were responders (week 24).

| Type of response to INF α2a/RBV 2005–2013 | DAA Treatment 2017 | Gender | Age | Elastography (Kpa) | Cirrhosis, CPS | Viral subtype | Initial viral load (Log$_{10}$ IU/mL) | HCV RNA at week 12 (end of treatment) | HCV RNA at week 24 | Type of response to DAAs | SVR 12 |
|---|---|---|---|---|---|---|---|---|---|---|---|
| Null response | OBV/PTV/r/DSV | M | 59 | 7.3 | No | 1b | 6.5 | Not detected | 1.62Log$_{10}$ UI/mL | Relapse | No |
| Breakthrough | OBV/PTV/r/DSV+RBV | M | 33 | 4.3 | No | 1a | 5.7 | 5.29Log$_{10}$ IU/mL | 5.10Log$_{10}$ UI/mL | Null response | No |
| Null response | OBV/PTV/r/DSV | F | 50 | 31.4 | Yes, CPS-A | 1b | 5.7 | *Detected | Not detected | Responder | Yes |
| Relapse | OBV/PTV/r/DSV | F | 54 | 5.3 | No | 1b | 5.3 | *Detected | Not detected | Responder | Yes |
| Relapse | OBV/PTV/r/DSV+RBV | F | 56 | 5.5 | No | *1a | 5.3 | *Detected | Detected/ Not detected at week 36[a] | Responder | Yes |
| Null response | OBV/PTV/r/DSV | M | 50 | 5.5 | No | 1b | 5.4 | *Detected | Not detected | Responder | Yes |
| Relapse | OBV/PTV/r/DSV | F | 52 | 11 | No | 1b | 4.3 | *Detected | Not detected | Responder | Yes |
| Null response | SOF/LDV | F | 52 | 4.3 | No | 1a | 6.0 | *Detected | Not detected | Responder | Yes |
| Null response | SOF/LDV+RBV | F | 52 | 19.8 | Yes, CPS-A | 1b | 6.4 | *Detected | Not detected | Responder | Yes |
| Null response | SOF/LDV+RBV | F | 68 | 22.3 | Yes, CPS-A | 1a | 5.5 | *Detected | Not detected | Responder | Yes |

**Notes.**

*This patient had a HCV viral load below the LLOQ but above of LLOD at week 12 (end of treatment).

[a]This patient had a positive qualitative test at week 12 (end of treatment) and at week 24 (12 weeks post-treatment), that became undetectable at week 36 (24 weeks post-treatment).

M, Male; F, Female; Kpa, Kilopascals; CPS, Child Pugh Score; LLOQ, Lower limit of quantification (1.39 Log$_{10}$ IU/mL); LLOD, Lower limit of detection (1.08Log$_{10}$ IU/mL); SVR12, sustained virological response for DAA treatments..

**Table 6  Biochemical parameters and clinical data, pre-/post-DAA treatments.**

| | OBV/PTV/r/DSV ± RBV | | | SOF/LDV ± RBV | | |
|---|---|---|---|---|---|---|
| | Pre-treatment | Post-treatment[*] | p | Pre-treatment | Post-treatment[*] | p |
| *Laboratory parameters, average (95% CI)* | | | | | | |
| Total bilirubin, mg/dL | 1.1 (0.9–1.3) | 1.1 (0.86–1.3) | 0.120 | 1.4 (1.2–1.8) | 1.6 (0.8–2.44) | 0.750 |
| Albumin, g/dL | 3.8 (3.6–3.9) | 4.1 (4.02–4.26) | <0.001 | 3.4 (3.3–3.6) | 3.8 (3.6–4.0) | <0.001 |
| Hemoglobin, g/dL | 14.5 (14.1–15.0) | 14.6 (14.1–15.1) | 0.891 | 14.1 (13.6–14.6) | 14.3 (13.8–14.8) | 0.271 |
| INR | 1.05 (1.01–1.08) | 1.10 (1.07–1.14) | <0.001 | 1.12 (1.08–1.17) | 1.19 (1.13–1.24) | 0.004 |
| ALT, IU/Lt[**] | 65 (52–79) | 22 (19.6–25.2) | <0.001 | 72 (59–84) | 28 (22.3–34) | <0.001 |
| AST, IU/Lt[**] | 64 (52–76) | 28 (25.4–31) | <0.001 | 86 (71–101) | 38 (33.9–42.6) | <0.001 |
| Platelets × 1000/mm$^3$ | 159 (134–185) | 166 (140–191) | 0.072 | 114 (96–133) | 117 (98–136) | 0.221 |
| AFP, ng/mL | 10 (6.32–13.7) | 4.7 (3.35–6.22) | <0.001 | 16 (10.1–22.3) | 8.2 (2.29–14.2) | <0.001 |
| *Child- Pugh- Score (CPS), n (%)[***]* | | | | | | |
| A (5–6 points) | 18 (95.0) | 19 (100) | | 16 (50.0) | 27 (84.4) | |
| B (7–9 points) | 1 (5.0) | 0 (0.0) | | 14 (43.8) | 4 (12.5) | |
| C (10–15 points) | 0 (0.0) | 0 (0.0) | | 2 (6.2) | 1 (3.1) | |
| Total cirrhosis | 19 (100) | 19 (100) | 1.00 | 32 (100) | 32 (100) | 0.012 |
| *Serological fibrosis markers* | | | | | | |
| Cirrhotic patients | | | | | | |
| FIB-4 | 4.5 (3.0–6.0) | 3.2 (1.9–4.5) | <0.025 | 7.6 (6.3–8.8) | 5.4 (4.2–6.6) | <0.001 |
| APRI | 2.3 (1.4–3.1) | 0.9 (0.5–1.3) | <0.001 | 3.6 (2.6–4.8) | 1.7 (1.4–2.1) | <0.001 |
| Non cirrhotic, CHC patients | | | | | | |
| FIB-4 | 2.0 (1.3–2.7) | 1.5 (1.1–1.9) | 0.05 | 2.3 (1.4–3.1) | 1.8 (1.2–2.5) | 0.03 |
| APRI | 0.9(0.5–1.4) | 0.4 (0.3–0.56) | 0.007 | 1.2 (0.6–1.7) | 0.5 (0.3–0.7) | 0.004 |

**Notes.**

INR, International Normalization Ratio (INR) of prothrombin time; ALT, alanine aminotransferase; AST, aspartate aminotransferase; AFP, Alpha-fetoprotein.

[*]Values post-treatment were measured 12 weeks after treatment end (at week 24).

[**]Normal reference values were ALT =10-45 IU/L and AST 10-43 IU/L.

[***]Only patients with cirrhosis are displayed in this section: 19 with OBV/PTV/r/DSV ± RBV and 32 with SOF/LDV ± RBV. FIB-4 cutoff value <1.45 corresponds to no hepatic fibrosis, >3.25 corresponds to F4 cirrhosis. APRI cutoff value <0.5 corresponds to no hepatic fibrosis, >1.5 corresponds to F4 cirrhosis.

# DISCUSSION

SVR is attained only in about half of CHC patients treated with pegIFNα+RBV but in over 90% of those treated with DAAs. Widespread access to DAAs was initially delayed by drug costs and accessibility, especially in some world regions that harbor most of the CHC patients (*World Health Organization, 2017*). The switch to the new treatments in those regions has been slow and relies importantly on public health systems. Mexico and Brazil are the countries in Latin America with the highest rates of cirrhosis, related to alcoholism and CHC (*Mendez-Sanchez et al., 2018*). Mexican public health institutions, such as IMSS, included DAAs in their list of essential medicines in June 2017, initially treating patients who had failed pegIFNα2a+RBV. This is the first report of the outcome of a Mexican cohort treated this way and followed long-term.

We found 36% SVR with the initial pegIFNα2a+RBV regimen (Table 2), similar to previous reports in Mexico of 32.5% (*Sandoval-Ramirez et al., 2015*). Fifty one percent of the non-cirrhotic patients, and 68% of those with low baseline viral load achieved SVR with

pegIFNα2a+RBV (Table 3), similar to other publications (*Enomoto & Nishiguchi, 2015*; *Naing et al., 2015*). Diabetes did not associate with failure of pegIFNα2a+RBV treatment: 34.2% of diabetic patients achieved SVR (Table 3), similar to SVR of all the cohort. The original cohort included equivalent amounts of patients with HCV subtypes 1a and 1b (Table 1), which had similar SVR rates with pegIFNα2a+RBV (Table 2).

Not all patients that displayed some initial response to pegIFNα2a+RBV (RVR, cEVR or pEVR) achieved SVR: 14.6% (38/261) relapsed (Table S1), comparable to previous reports of 16% and 14.4% relapse (15,17). Thus, RVR, cEVR, pEVR were not predictors of SVR or relapse with pegIFNα2a+RBV (Table 3). The low SVR rate with this treatment was likely associated to the high frequency of cirrhosis and the high viral load present in our initial cohort (Table 1). This was expected, as virus elimination is difficult with pegIFNα2a+RBV when the viral load is high, in particular for genotype 1 (*Enomoto & Nishiguchi, 2015*).

Of the 167 non-responder patients to pegIFNα2a+RBV, only 83 returned for DAA treatment and all of them concluded therapy with either *OBV/PTV/r/DSV ± RBV* or *SOF/LDV ± RBV*. DAA treatment happened on average 7 years after pegIFNα2a+RBV treatment, thus the sub-cohort with DAA treatment was older (56.2 *vs* 49.5 years), had a higher cirrhosis rate (60% *vs* 29%) and a lower basal average viral load, when they received DAAs than at pegIFNα2a+RBV treatment.

Women were 74.6% of the patients treated with DAAs, which is a high percentage compared to other DAA real-world studies that report 35% (*Perello et al., 2017*), 45% (*Flisiak et al., 2016*) 54% (*Mendizabal et al., 2017*), and 44.4% (*Holzmann et al., 2018*) females (the last two in Latin America). The high proportion of women in our cohort likely reflects that in Mexico an important risk factor for CHC is the history of blood transfusion prior to 1995 (*Lopez-Colombo et al., 2014*) likely during obstetric or gynecological procedures. Females had a higher mean age than men in our DAA sub-cohort (Table 4).

SVR in the group treated with *OBV/PTV/r/DSV ± RBV* was 94.8%, similar to other studies that have reported 96% (*Welzel et al., 2017*), 99% (MALACHITE II trial) (*Dore et al., 2016*), and 98.7% (AMBER study) (*Flisiak et al., 2016*), in patients with previous treatment; or 96.8% (*Mendizabal et al., 2017*) and 96.2% (*Perello et al., 2017*) including both treatment-naïve and previously treated patients respectively. Two patients presented failure to *OBV/PTV/r/DSV ± RBV*, that did not associate with any of the studied factors (Table S2), similar to previous reports (*Flisiak et al., 2016*). However non-responders were 2/10 men in contrast to 0/29 women, and 2/21 patients without cirrhosis, in contrast to 0/18 patients with cirrhosis (Table S2).

SVR in the group treated with *SOF/LDV ± RBV* was 100%, similar to two multicenter studies that have reported 95.8%, and 92.5 to 100%, respectively (*Calleja et al., 2017*; *Terrault et al., 2016*) in patients with previous treatment; 99% in cirrhotic patients with viral subtype 1b (*Ogawa et al., 2017*), and a meta-analysis reported ≥ 95% (*Rezaee-Zavareh et al., 2017*). Our *SOF/LDV ± RBV* group had 72.7% of cirrhotic patients and viral subtype 1a predominated over 1b (Table 2). Male gender (*Ogawa et al., 2017*), basal albumin <3.5 g/dL, and basal total bilirubin >2.0 mg/dL have been associated with failure to *SOF/LDV ± RBV* treatment (*Calleja et al., 2017*; *Terrault et al., 2016*). In contrast, in our study 75%

of patients with *SOF/LDV* ± *RBV* were women; 47.8% had basal albumin >3.5 g/dL and 68% had basal total bilirubin <2.0 mg/dL, likely favoring SVR (Table 4).

Biochemical improvement was observed after both DAA treatments, particularly 10% increase in albumin levels, up to 3-fold decrease in ALT and AST levels, and no decrease in hemoglobin despite 79% of patients with SOF/LDV receiving RBV. In agreement with lack of change in hemoglobin, indirect bilirubin did not increase with DAAs either (Table 6). An AFP concentration above 10 ng/mL was found in 28.9% of patients pretreatment and the concentrations decreased by 50% after DAA treatment (Table 5). Only one patient (4.16%) showed an increase in AFP levels after treatment, in contrast to 22.9% found in another study (*Fouad et al., 2019*). Biochemical improvement translated into better CPS: 12 cirrhotic patients improved the CPS post-DAAs (Table 5). Other studies report changes in CPS at 36 weeks (*El-Sherif et al., 2018*), and at 6 months (*Essa et al., 2019*) post DAAs.

Ninety eight percent of our patients eradicated HCV after 12 weeks of treatment, but we detected residual HCV RNA at treatment end (12 weeks), in 8 patients that became undetectable in the following months (Table S2), A study reported that normalization of albumin, AST, and ALT levels after DAA treatment is associated with the restoration of immune activity (*Kostadinova et al., 2018*), suggesting that the immune response may clear the residual virus in the following weeks.

Still, not all patients showed biochemical improvement. For example, elevated ALT persisted post-treatment in 8 (9.8%) of patients, corresponding to patients with cirrhosis. This suggests the persistence of chronic liver inflammation despite SVR with DAAs in some cirrhotic patients, as has been observed (*Enomoto et al., 2018*).

The serological fibrosis markers FIB-4 and APRI showed a significant decrease after DAA treatment ($p < 0.05$) (Table 5). However, their values suggest that cirrhosis and liver fibrosis were not eliminated by DAA treatments (Table 5). A recent study with non-cirrhotic patients reported that APRI and FIB-4 rates decrease rapidly and steadily from week 2 to week 12 post-DAA treatment (*Hsu et al., 2019*). Another study reported a decrease in transient elastography 18 months after treatment, but the authors discuss that it remains to be examined whether this indicates a true regression of fibrosis or simply the resolution of chronic liver inflammation (*Bachofner et al., 2017*). A study with liver biopsies of patients that reached SVR, reported a decrease in the Knodell inflammatory score, and did not observe short-term improvement in fibrosis post-DAA treatment (41 ± 20 weeks after treatment end) (*Enomoto et al., 2018*). Thus, our observation on FIB-4 and APRI decrease wit treatment, could be due more to an improvement in chronic liver inflammation, which is supported by the significant decrease in liver transaminases (Table 5).

The biochemical and hepatic-fibrosis characterization of patients treated with DAAs, is instrumental to understand details beyond SVR, in particular related to liver inflammation and its contribution to long-term outcomes, like HCC. Several studies report that patients with cirrhosis remain at risk of HCC despite SVR, irrespective of the treatment (even with DAAs) (*Chinchilla-Lopez et al., 2017*; *Waziry et al., 2017*). In contrast, in patients without cirrhosis, a decrease in liver inflammation reduces the risk of cirrhosis and HCC (*Hsu et al., 2019*). This suggests that the best window for DAA treatment is before the onset of cirrhosis.

## CONCLUSIONS

DAA treatment showed good tolerability and safety, as well as excellent SVR rates in Mexican patients who had been unsuccessfully treated with pegIFNα2a+RBV several years earlier. Child-Pugh-Score improved in some patients with cirrhosis. Treatment with DAA did not correct cirrhosis, but FIB-4 and APRI suggest a reduction in chronic liver inflammation.

### Funding
Costs of medical attention and diagnosis were covered by Instituto Mexicano del Seguro Social. Julio Reyes-Leyva received a research fellowship from Fundación IMSS A.C., Mexico. There was no additional external funding received for this study. The funders had no role in study design, data collection and analysis, decision to publish, or preparation of the manuscript.

### Grant Disclosures
The following grant information was disclosed by the authors:
Instituto Mexicano del Seguro Social.
Fundación IMSS A.C., Mexico.

### Competing Interests
The authors declare there are no competing interests.

### Author Contributions
- Daniel Melendez Mena conceived and designed the experiments, performed the experiments, prepared figures and/or tables, authored or reviewed drafts of the paper, and approved the final draft.
- Miguel Angel Mendoza-Torres conceived and designed the experiments, performed the experiments, prepared figures and/or tables, and approved the final draft.
- Virginia Sedeno Monge conceived and designed the experiments, analyzed the data, prepared figures and/or tables, authored or reviewed drafts of the paper, and approved the final draft.
- Víctor Hugo García y García performed the experiments, analyzed the data, prepared figures and/or tables, and approved the final draft.
- Elain Rivera-García, Laura Sánchez-Reza, María del Carmen Baxin Domínguez and Belinda Guzmán-Flores performed the experiments, prepared figures and/or tables, and approved the final draft.
- José Manuel Coronel Espinoza, Iván Galindo-Santiago and Juan Carlos Flores-Alonso performed the experiments, authored or reviewed drafts of the paper, and approved the final draft.
- Verónica Vallejo-Ruiz performed the experiments, analyzed the data, prepared figures and/or tables, authored or reviewed drafts of the paper, and approved the final draft.

- Paulina Cortes-Hernandez, Francisca Sosa-Jurado and Gerardo Santos-López conceived and designed the experiments, analyzed the data, prepared figures and/or tables, authored or reviewed drafts of the paper, and approved the final draft.
- Julio Reyes-Leyva conceived and designed the experiments, prepared figures and/or tables, authored or reviewed drafts of the paper, and approved the final draft.

## Human Ethics

The following information was supplied relating to ethical approvals (i.e., approving body and any reference numbers):

The study was performed in accordance with ethical regulations and approved by the local committee of research and ethics No. 2101, IMSS (Registry numbers R-2004-2101-008, R-2008-2101-10, and R-2019-2101-001). Informed consent was obtained from all patients prior to any of the treatments.

## Data Availability

The raw data is available in the Supplemental Files.

## Supplemental Information

Supplemental information for this article can be found online at http://dx.doi.org/10.7717/peerj.12051#supplemental-information.

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
