# Peer review of "Effectiveness, tolerability and safety of Direct Acting Antivirals in Mexican individuals with Hepatitis C virus genotype-1 and previous pegylated interferon and ribavirin therapy"

_PeerJ, doi:10.7717/peerj.12051_

## Round 0.1 · original submission · Minor Revisions

Please review carefully all comments provided by the reviewers and addressed them in detail.

·

Basic reporting

Although the manuscript is decent enough in clarity, it needs to be revised accordingly to further enhance the clarity and unambiguous. Also, I strongly believe that the English language should be improved to ensure that an international audience can clearly understand your text. Thus, I suggest the authors have a colleague who is proficient in English and familiar with the subject matter review the manuscript, or contact a professional editing service.

At times, authors have used the old citations (around 10 years old) where a lot of advances and improvements have taken place within the literature.

Table and figure titles and legends have to be revised according to the context being reported/illustrated (especially look for grammar and comprehension)

Experimental design

This is original research within the aims and scope of the journal, where the research question was well-defined that fills an identified knowledge gap. Rigorous investigation has been performed to a technical and ethical standard. Although methods were described sufficiently, I suggest authors specifically mention the type of the study design in both abstract and the main text. Also, I am wondering about the gap (in years) between two sub-cohorts i.e., what about years 2015 and 2016 that are missing from both the sub-groups.

Validity of the findings

Meaningful replication encouraged where rationale & benefit to literature was clearly stated. All underlying data have been provided, which are robust. The data on which the conclusions were based are made available. However, I believe the conclusions should be well-stated further based upon the results.

Additional comments

All the general comments can be found as annotations or comments in the manuscript pdf file itself. I recommend the authors to consider the comments provided while revising the manuscript.

Reviewer 2 ·

Basic reporting

Melendez Mena et al. report the effect of Hepatitis C treatment with Direct Acting Antivirals (DDAs) in a cohort of 83 Mexican patients who had not responded to IFN/ribavirin treatments. The study focuses on patients with genotype 1 strains.
The results show that while IFN/ribavirin treatment had failed in more that 60% of cases, DDAs resulted in sustained virological response in 97% of patients, with marked improvement of liver function and inflammation parameters.
This study is well conducted and the manuscript is well written, although the only novelty compared to the existing literature is the Mexican context.
I only have minor comments, mostly regarding the presentation of the data:
1) Line 56. Why “CTP”? Perhaps the authors mean “CPS”
2) Line 57. Why “despite”? All parameters improve and so it is expected that transaminases also decrease.
3) Line 62. CHC is not defined. Also, many acronyms are defined only in the abstract and that may confuse the uninitiated reader, since this is a very acronym-heavy manuscript. I suggest that the authors make sure all the acronyms are defined in the main text and, perhaps, that they are used more sparingly.
4) Sagnelli et al., 2013 is not the right reference here, since it does not mention anything about worldwide mortality.
5) Line 213-214 – It is not clear here how fibrosis was measured and I cannot see the results of these measurement in table 5 or anywhere else in the manuscript. Yet, lack of change in fibrosis is mentioned in the abstract and throughout the manuscript. The author should explain this point more clearly.
6) Line 219. Could elevated bilirubinemia due to ribavirin treatment or does it occur also in –RBV patients? Could the author comment on this?
7) Line 221. Here and throughout the manuscript, the authors describe the treatment groups as +/-RBV, but they never explain how the cohort was divided for +RBV and –RBV treatment, why, and how many f patients were in each group. The authors should describe the experimental design in more details in M&M and clearly define the number of patient in each group.
8) The first part of the discussion, especially lines 241 and 247 is too detailed and the “data not shown” should be presented and described in Results.
9) Line 391. “Installation” does not seem to be the right word. Perhaps “onset” or “development”?
10) Fig. 1 is not referenced in the text and not described anywhere. It looks like the data of Fig. 1 are also shown in the various Tables, and I would suggest to remove it.
11) I suggest to move Supp table 1,2 and 4 to the main text, because these are very relevant data for the study.

Experimental design

The experimental design is adequate and the research is meaningful. Please see some of my minor comments under Basic Reporting asking for clarification of some aspects of the experimental design.

Validity of the findings

The finding are valid and relevant and the conclusions are well supported by the data.

Additional comments

None

---

## Round 0.2 · accepted · Accept

It seems that the issues raised by the reviewers have been resolved.
Thank you for your work.

Reviewer 2 ·

Basic reporting

Melendez Mena et al. report the effect of Hepatitis C treatment with Direct Acting Antivirals (DDAs) in a cohort of 83 Mexican patients who had not responded to IFN/ribavirin treatments.
This is the first revision of a manuscript that I have previously reviewed. The authors have addressed satisfactorily all my previous suggestions and I don’t have any further comment.

Experimental design

No comments

Validity of the findings

No comments

Additional comments

No comments